# RoMeO: Robust Metric Visual Odometry

## Abstract

Visual odometry (VO) aims to estimate camera poses from visual inputs — a fundamental building block for many applications such as VR/AR and robotics. This work focuses on *monocular RGB VO* where camera poses are directly estimated from a monocular RGB video *without IMU or 3D sensors*. Existing approaches lack robustness under this challenging scenario and fail to generalize to unseen data (especially outdoors); they also cannot recover metric-scale poses. Several methods have attempted to address these problems with priors from predicted depth. However, especially on unseen data, depth prediction noise can drastically degrade performance. We propose *Robust Metric Visual Odometry (RoMeO)*, the first method that can leverage (noisy) depth priors to enable robust VO and recover metric scale poses. RoMeO incorporates both pre-trained monocular metric depth and multi-view stereo (MVS) models to recover metric-scale, simplify correspondence search, provide better initialization and regularize optimization. Effective strategies ensure the efficiency and the robustness to prior noise. RoMeO advances the state-of-the-art (SOTA) by a large margin *across 6 diverse datasets covering both indoor and outdoor scenes*. Compared to the current SOTA DPVO, RoMeO reduces the relative (align the trajectory scale with GT) and absolute trajectory errors on average by $55.2\%$ and $77.8\%$ respectively (Fig. 1). The performance gain also transfers to the full SLAM pipeline (with global BA & loop closure). Code will be released upon acceptance.

## 1 Introduction

Visual Odometry (VO) estimates the sensor pose from visual signals. It is the core problem of many applications such as mapping, robot navigation, and autonomous driving. This work focuses on *monocular RGB VO*, where the input is only a monocular RGB video and no information from a 3D sensor or inertial measurement unit (IMU) is available.

Classical methods rely on hand-crafted features and explicit geometric optimizations to estimate poses. The geometric optimization is executed either directly on pixel intensities (Engel et al., 2014; 2017; Zubizarreta et al., 2020), or indirectly on matches between keypoints (Mur-Artal et al., 2015; Rosinol et al., 2020). Due to the instability of hand-crafted features, classical methods suffer from frequent tracking failures on challenging data with large motions or extreme weather.

Learning-based approaches (Wang et al., 2021; Teed & Deng, 2021; Teed et al., 2024) train end-to-end systems for both correspondence search and pose optimization, which minimizes tracking failures. However, these approaches have limited robustness, i.e., they generalize poorly on zero-shot data, where classical methods often perform better when they do not fail. Moreover, both types of methods lack mechanisms to recover metric-scale trajectories without 3D sensors or IMU.

A common reason for these drawbacks is the lack of priors. As a result, the joint pose-depth optimization suffers from local optima and cannot recover the metric-scale. Though previous works have explored the use of predicted depth (Yang et al., 2020; Yin et al., 2023) to VO, the depth prediction noise limits their robustness, making them often perform worse on challenging datasets than pure correspondence-based SOTA (Teed & Deng, 2021; Teed et al., 2024). To this end, we propose *Robust Metric Visual Odometry (RoMeO)*, a novel method that can leverage potentially noisy priors from pre-trained depth models to enhance VO robustness and recover metric-scale poses.

RoMeO initializes VO with light-weight monocular metric depth models to recover metric scale poses. Robust depth-guided bundle adjustment adaptively detects accurate monocular and MVS

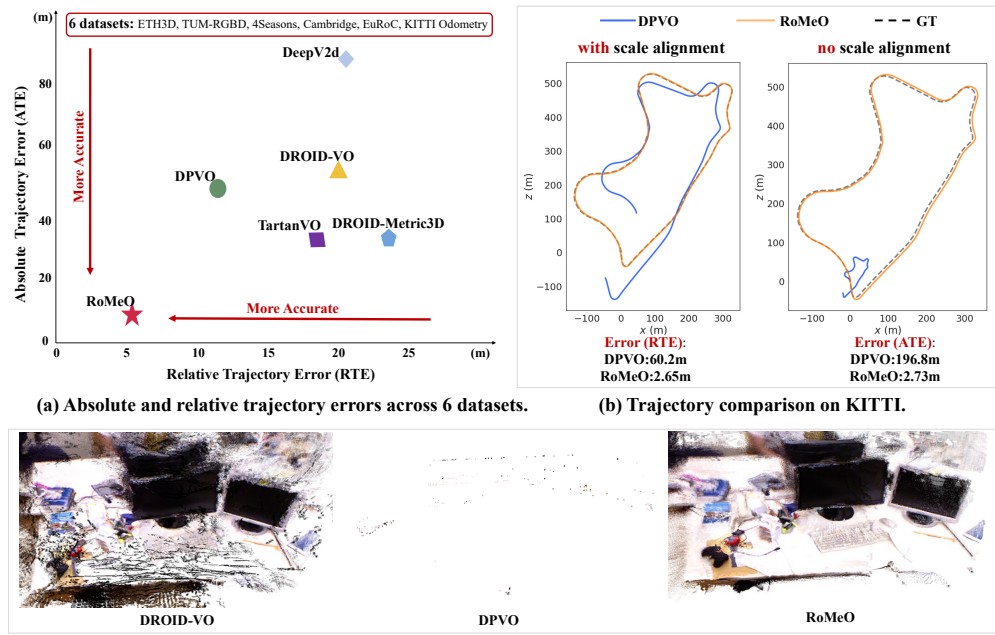

(a) Absolute and relative trajectory errors across 6 datasets.

(b) Trajectory comparison on KITTI.

(c) Point cloud comparison. DPVO performs sparse reconstruction to accelerate VO.

Figure 1: **Teaser**. We propose RoMeO, a robust metric-scale VO system for monocular RGB videos. RoMeO effectively introduces priors from pre-trained depth estimation models to standard flow-based VO systems. (a) On 6 diverse datasets covering both indoor and outdoor scenes, RoMeO consistently and significantly improves the performance, both in terms of the trajectory shape (relative trajectory error (RTE)) and scale (absolute trajectory error (ATE)). (b) RoMeO trajectories closely align with the ground truth, even without scale alignment. (c) RoMeO enables dense 3D reconstructions with a much higher quality compared to previous SOTA.

priors, and uses them to regularize pose-depth optimization. Effective conditions are introduced to avoid poor MVS priors generated by inaccurate poses, low multi-view overlaps, or insufficient motions. Noise augmented training adapts the flow estimation network to the depth-enhanced inputs, which maximizes the accuracy while maintaining the robustness to depth prediction noise.

As shown in Fig. 1 (a), RoMeO significantly advances the SOTA across 6 diverse benchmarks covering both indoor and outdoor scenes. Compared to the current SOTA DPVO (Teed et al., 2024), RoMeO achieves 55.2% and 77.8% reduction respectively for relative and absolute trajectory errors, i.e., both the trajectory shape and scale are significantly improved (Fig. 1 (b)). Unlike previous depth-based approaches (Yin et al., 2023) that hurt the VO accuracy on challenging data, the performance improvement of RoMeO is consistent across the board. RoMeO also enables dense and much more accurate 3D reconstructions (Fig. 1 (c)). These improvements also propagate to the full SLAM system with global BA enabled. Extensive analyses validate the individual RoMeO components.

**Contributions:** (1) We devise a novel learning-based VO-system that generalizes to unseen data and outperforms SOTA by a large margin across 6 benchmarks. (2) We analyze both monocular and MVS depth models and provide solutions to obtain high-quality priors while maintaining efficiency. (3) We introduce robust depth guidance into flow estimation and bundle adjustments, which significantly improves the accuracy even under noisy depth priors.

## 2 RELATED WORK

**Visual odometry.** Practical VO systems may rely on inputs beyond a monocular video, for example, IMU in visual-inertial odometry (Forster et al., 2015), sensor depth in RGB-D VO (Whelan et al., 2013; Teed & Deng, 2021; Handa et al., 2014), and multi-view sensors in stereo VO (Wang et al., 2017; Engel et al., 2014). The focus of this work is VO from only monocular RGB video.

Figure 2: **Overview**. RoMeO initializes each frame using monocular metric depth models. MVS models are used to further refine intermediate BA depth. Besides replacing the initial/intermediate depth, monocular and MVS depth priors are also added into the regularization terms of BA, with adaptive conditions to filter noisy depth priors and enable effective MVS prediction. Noise augmented training is used to adapt the flow network to depth-enhanced inputs, which maximizes the accuracy while maintaining the robustness to prior noise.

Classical methods use hand-crafted features to find correspondences and perform joint pose-depth optimization. Indirect approaches (Mur-Artal et al., 2015; Rosinol et al., 2020) find correspondences through feature matching and then minimize the projection error on the correspondences. Direct approaches (Engel et al., 2014; 2017; Zubizarreta et al., 2020) minimize the photometric error directly without feature matching. A common drawback of classical methods is the frequent tracking failure on challenging data with large motions or extreme weather.

Learning-based methods train differentiable systems end-to-end on labeled data to avoid tracking failures. Teed & Deng (2018) proposed a differentiable structure-from-motion architecture that alternates between motion and depth estimation. Wang et al. (2021) incorporated camera intrinsics into VO and pioneered training on the large-scale synthetic dataset TartanAir (Wang et al., 2020). Teed & Deng (2021) significantly improved prior works by introducing RAFT (Teed & Deng, 2020) into VO systems and designing a differentiable bundle adjustment layer for pose-depth joint optimization. Teed et al. (2024) sped up Droid-SLAM by sparse flow estimation and optimization. The major drawback of learning-based approaches is the limited robustness to zero-shot data, especially for outdoor scenes with challenging motion and dynamic objects. As a result, they can perform worse than classical methods which fail less in such cases. Meanwhile, existing approaches cannot recover metric-scale poses without 3D sensors (depth, stereo, or IMU). Though several works (Tateno et al., 2017; Yang et al., 2020; Yin et al., 2023) tried to introduce predicted depth to VO, they suffer from performance drop due to the depth prediction noise, especially on zero-shot data. RoMeO improves VO by introducing depth priors from pre-trained monocular and MVS models to both the initialization and the iterative optimization of VO. Effective strategies are proposed to ensure the robustness of RoMeO under (severe) prior noise, which enables consistent and significant error reduction across diverse zero-shot data.

**Depth estimation.** *Monocular depth* estimation aims to recover the depth from a single input image. Recent methods (Ranftl et al., 2020; 2021; Bhat et al., 2023; Yin et al., 2023; Yang et al., 2024) have trained models on large-scale data to enable zero-shot monocular depth estimation. *Multi-view stereo (MVS)* methods (Yao et al., 2018; Bae et al., 2022; Cao et al., 2022) estimate depth from posed multi-view images. With accurate poses for the input images, they can recover more consistent and accurate depth than monocular methods. RoMeO leverages both monocular and MVS models to better initialize and regularize VO.

## 3 METHOD

As shown in Fig. 2, RoMeO maintains a frame graph on-the-fly during VO, which contains a sliding window of keyframes. Every new frame is initialized with an identity relative pose and the predicted monocular *metric* depth from a pre-trained model. Given the current frame graph, RoMeO jointly refines the poses and depth by alternating between *flow estimation* and *differentiable bundle adjustment (BA)*, with robust guidance from both monocular and MVS depth.

Inspired by the recent SOTA (Teed & Deng, 2021; Teed et al., 2024), each *flow estimation* step first obtains an initial optical flow by projecting the pixels of each keyframe to the others using camera intrinsics and the current pose and depth. This initial flow is then fed to a flow network that predicts a *residual flow* and a confidence map. RoMeO leverages depth priors to obtain more accurate initial flow, which simplifies residual flow prediction. Any flow model can in principle be used in this step. Due to the wide adoption and the ability to perform dense reconstructions, we use a RAFT-style architecture following (Teed & Deng, 2021). After flow estimation, 2 iterations of *BA* are performed to optimize the pose and depth. RoMeO leverages monocular metric depth to initialize BA such that it can recover metric scale poses without 3D sensor or IMU. Both monocular and MVS depth are further used to effectively regularize the BA objective. This process is repeated 6 times, resulting in 6 flow estimation steps and 12 BA iterations for each new keyframe. Pose-only BA is performed for non-keyframes, with monocular depth initialization.

## 3.1 ROBUST DEPTH-GUIDED BUNDLE ADJUSTMENT

RoMeO guides BA with both monocular and MVS depth. Besides using them to replace initial/intermediate BA depth, it embeds them into the BA objective below:

$$E(\mathbf{G}, \mathbf{d}) = \sum_{(i,j) \in \varepsilon} (\|\mathbf{p}_{ij}^* - \Pi_c(\mathbf{G}_{ij} \circ \Pi_c^{-1}(\mathbf{p}_i, \mathbf{d}_i))\|_{\Sigma_{ij}}^2 + C_i \lambda \|\mathbf{d}_i - \mathbf{d}_i^*\|_{\Sigma_{ij}}^2), \qquad (1)$$

where $\mathbf{G}$ and $\mathbf{d}$ are the optimized poses and depth, and $\epsilon$ is the current frame graph. The first term is the standard BA objective (Teed & Deng, 2021) where $\mathbf{p}_{ij}^* \in \mathbb{R}^{H \times W \times 2}$ denotes the image coordinates when we project the pixels from frame $i$ to $j$ *with the estimated flow*. $\Pi_c(\mathbf{G}_{ij} \circ \Pi^{-1}(\mathbf{p}_i, \mathbf{d}_i))$ corresponds to the image coordinates when we project the pixels from frame $i$ to $j$ *with the current depth $\mathbf{d}_i$ of frame $i$, the relative pose $\mathbf{G}_{ij}$ between frames $i$ and $j$*. $\Pi_c$ is the world-to-camera projection. Intuitively, the first term encourages the consistency between flow-based and geometric-projection-based correspondences. The second term is a robust depth regularization term where we encourage the BA-optimized depth map $\mathbf{d}_i \in \mathbb{R}^{H \times W}$ to be close to the predicted monocular/MVS depth $\mathbf{d}_i^*$. $\lambda = 0.05$ is the penalty weight for regularization. The condition weight $C_i \in \{0, 1\}$ turns off depth regularization when severe noise exists in $\mathbf{d}_i^*$. $\| \cdot \|_{\Sigma_{ij}}$ is the Mahalanobis distance which weights the error terms based on the confidence $\mathbf{w}_{ij} \in [0, 1]^{H \times W}$ of the flow estimation.

RoMeO automatically determines the value of $C_i$, which is the key to making depth-guided BA robust to prior noise. During the construction of the initial frame graph containing the first 12 keyframes of the scene, we do not enable depth regularization ($C_i = 0$). After the initial frame graph has been built and the pose-depth optimization is completed on this graph, we compute the average photometric error from the latest keyframe $i$ to the connected frames in the graph:

$$\eta(i) = \sum_{(i,j) \in \varepsilon'} \|\mathbf{c}_i(\mathbf{p}_i) - \mathbf{c}_j(\Pi_c(\mathbf{G}_{ij} \circ \Pi_c^{-1}(\mathbf{p}_i, \mathbf{d}_i)))\|_{\Sigma_{ij}}^2, \qquad (2)$$

where $\mathbf{c}_i(\mathbf{p}_i) \in \mathbb{R}^{H \times W \times 3}$ returns the colors of the pixels in frame $i$. Eq. 2 essentially computes the color difference between the pixels in frame $i$ with their corresponding pixels in frame $j$ where the correspondences are obtained by the same pose-depth re-projection in Eq. 1. When the depth of frame $i$ is accurate, the photometric error $\eta(i)$ should be small. We denote the error on the latest keyframe as $\eta_{\text{init}}$.

For every new keyframe in the future, we compute the same photometric error $\eta'$ using Eq. 2, where the depth is the initial monocular depth, and the pose is obtained from the first BA iteration. We only set $C_i$ to 1 if $\eta' < \alpha \eta_{\text{init}}$, where $\alpha$ is a pre-defined constant. Intuitively, when this condition is violated, the predicted depth is potentially unreliable and should not be used to regularize BA. This strategy is robust to noise: it not only maintains the strong performance gain from accurate depth priors but also mitigates the negative impacts of severe depth noise. As a result, RoMeO achieves much higher zero-shot accuracy than current SOTA across a diverse set of datasets.

## 3.2 METRIC DEPTH PRIOR

The non-convex nature makes VO optimization highly sensitive to initialization. State-of-the-art methods often initialize the depth of a new frame to a constant (e.g. 1m) (Teed & Deng, 2021) for

all pixels, which might be acceptable for indoor scenes where the depth variation is small. However, it is unsuitable for outdoor scenes where close and distant pixels can have highly different depths. Uniform initialization also hinders the recovery of metric-scale poses. To address both problems, we initialize the depth of each frame with the output of a pre-trained monocular *metric* depth model.

Table 1: **Different monocular depth models for initialization and depth guided BA.** Other strategies mentioned in later sections are not applied.

| Model | KITTI Odometry | | TUM-RGBD | |
|---|---|---|---|---|
| | RTE (m) / ATE (m) | FPS | RTE (m) / ATE (m) | FPS |
| No depth prior | 47.53/137.33 | **5.33** | 0.116/0.551 | **10.77** |
| DepthAnythingV2-Small | 11.15/18.78 | 4.19 | 0.104/0.273 | 7.56 |
| DepthAnythingV2-Large | 7.97/13.47 | 1.59 | 0.107/0.456 | 4.20 |
| Metric3DV2-Small | 5.71/8.74 | 4.08 | 0.104/0.235 | 6.41 |
| Metric3DV2-Large | **4.08/5.73** | 1.70 | 0.105/0.247 | 2.39 |
| DPT-Hybrid | 4.25/8.54 | 3.96 | **0.098/0.205** | 8.55 |

Though more recent methods (Hu et al., 2024; Yang et al., 2024) with large pre-trained models are available, they introduce severe overheads to the VO system and hinder the practicality. To minimize overhead while maintaining robustness, we use the lightweight DPT-Hybrid (Ranftl et al., 2021) with the provided scale and shift parameters to obtain the metric depth. Tab. 1 shows the accuracy and speed of VO using different metric depth models for initialization and BA regularization. We use 1 indoor (KITTI) and 1 outdoor (TUM-RGBD) datasets for analysis. Interestingly, though all depth models can provide reasonable error reduction, bigger models such as DepthAnythingV2-Large (Yang et al., 2024) and Metric3DV2-Large (Hu et al., 2024) introduce severe overheads to VO, and are not obviously better than DPT-Hybrid. We conjecture that this is because the BA of RoMeO will optimize the depth details and filter out noisy depth priors. Therefore, the key to practical monocular depth guidance in VO is a light-weight model that can provide rough but robust metric-scale depth.

### 3.3 MVS Prior

Creating effective MVS guidance in a VO system is non-trivial for three main reasons: (1) *Accurate MVS requires a reasonable amount of camera translations and rotations, and large overlapping areas between multiple views.* These conditions are not ensured by standard keyframe selection strategies. (2) *Accurate MVS needs accurate poses.* Naively adding MVS guidance at early iterations can hurt VO. (3) *MVS outputs often contain noise.*

To address the first issue, we denote the intermediate relative translation estimated by BA between the last 3 keyframes $(i-2, i-1, i)$ as $\mathbf{t}_{i-2,i-1}$ and $\mathbf{t}_{i-1,i}$, and only enable MVS guidance when

$$\|\mathbf{t}_{k-1,k}\| + \|\mathbf{t}_{k-2,k-1}\| > 0.1m \ \ \& \ \ \angle(\mathbf{t}_{k-2,k-1}, \mathbf{t}_{k-1,k}) \in [10°, 30°]. \tag{3}$$

This strategy can effectively filter keyframes with extreme motions or small cross-view overlaps.

For the second issue, an interesting observation is that, given effective monocular depth guidance, BA can return reasonably accurate intermediate poses within a small number of iterations. This is evident from Fig. 3 where we plot the magnitude of the residual flow over different BA iterations for the *KITTI Odometry* dataset (Geiger et al., 2012). The magnitude of RoMeO at early iterations is already lower than the final iteration of the baseline without any depth prior, which indicates better pose accuracy. Based on this observation, if E.q. 3 is satisfied after the 8-th BA iteration, RoMeO uses the intermediate poses and the most recent 3 keyframes to compute the MVS depth for the newest keyframe.

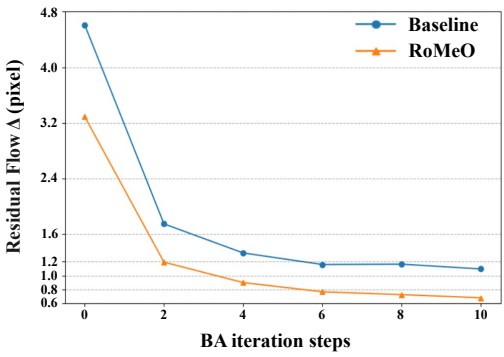

Figure 3: **Residual flow magnitudes in different BA iterations.**

We use pre-trained MaGNet(Bae et al., 2022) as the MVS estimator. The intermediate BA depth and $\mathbf{d}_i^*$ in E.q. 1 are then replaced with the MVS depth to improve the remaining flow generation and pose-depth optimization accuracy.

To address the third issue, we leverage the output confidence map $\mathbf{M} \in \mathbb{R}^{H \times W}$ of MVS, which indicates the probability that the prediction on each pixel is correct. Specifically, we ignore the

pixels corresponding to the lowest 20% of the values in M during future flow generation and BA iterations. Essentially, sparse BA is conducted on selected pixels with reliable depth.

These strategies not only improve the robustness of MVS guidance but also reduce the overhead.

### 3.4 Noise Augmented Training

The trainable parameters of RoMeO lie in the flow network. Besides benefiting BA optimization, accurate depth priors effectively simplifies flow estimation since the initial flow obtained by pose-depth based reprojection would be more accurate. Empirically we can train the flow network without depth priors, and insert depth priors during inference to achieve non-trivial performance improvements. However, this naive strategy cannot fully adapt the trainable parameters to the prior-enhanced inputs. Specifically, our prior-enhanced poses and depth lead to more accurate initial flow, which requires a much smaller residual flow during inference. Hence, we jointly train the flow network with the depth-enhanced inputs.

Since predicted depth can have (severe) noise on unseen data, we purposely maintain some noise during training to ensure the robustness to prior noise. Specifically, we first pre-train the flow network without any depth prior. Then, we fine-tune the pre-trained system with monocular depth initialization with a much smaller learning rate. MVS depth is not used during fine-tuning to ensure a reasonable amount of depth noise.

The fine-tuning is done on *TartanAir* (Wang et al., 2020), the same simulation dataset used for pre-training. There is a strong domain gap between simulation and real data, for example, some TartanAir scenes contain extreme depth values rarely seen in real-world data (e.g., objects that are $16km$ away). The monocular depth models often return heavily erroneous predictions in these cases, making fine-tuning ineffective. To address this problem, we align the scale and shift of the predicted depth to the ground truth when the relative prediction error exceeds 20%. This strategy minimizes the negative impact of domain shift, making the flow network adapt to more accurate depth in practice. Meanwhile, only aligning predictions with large errors ensures that the fine-tuning data still contain a reasonable amount of noise.

With effective noise handling strategies, the depth prior in RoMeO significantly enhances flow generation since the cross-view re-projection is much more accurate. As supported by Fig. 3, a strong depth prior leads to much smaller flow adjustment during iterative refinement, i.e., reliable correspondences are obtained much faster. A strong depth prior also helps BA to avoid bad local optima, making it easier to find accurate poses. These advantages create a *positive feedback loop*, where better depth leads to better flow and poses, which further enhance depth optimization.

## 4 Experiments

**Implementation.** RoMeO is implemented using a mixture of PyTorch and C++ following the official code of (Teed & Deng, 2021). To maximize the performance, we use separate depth models/hyperparameters for indoor and outdoor scenes. For depth initialization, we use DPT-Hybrid with the provided scale and shift hyperparameters (scale=0.000305, shift=0.1378 for indoor and scale=0.00006016, shift=0.00579 for outdoor) (DPT). For MVS guidance, we use the corresponding MaGNet (Bae et al., 2022) models for indoor and outdoor scenes. We set $\alpha$ in E.q. 2 to 1.75 and 1.5 respectively for outdoor and indoor scenes. Note that although the depth models/hyperparameters are different, we only train a *single* VO system and use it for both indoor and outdoor scenes. During RoMeO fine-tuning, we only use the outdoor DPT scale and shift to predict the initial depth, since TartanAir is mostly outdoor. The improper scale and shift will be re-aligned with GT anyway when they create large prediction errors. The initial learning rate of fine-tuning is reduced to 0.0001, with other setups the same as pre-training. Both stages of noise augmented training require roughly 7 days on 4 RTX-3090 GPUs.

**Data.** Previous monocular RGB VO systems are mostly evaluated on indoor scenes. We use 6 zero-shot datasets with diverse indoor and outdoor scenes to thoroughly evaluate the robustness of different algorithms:

- *KITTI Odometry* (Geiger et al., 2012): an outdoor self-driving dataset.

Table 2: **Visual Odometry Evaluation**. Results are reported as RTE (m) (with scale alignment) / ATE (m). Missing values (–) indicate that the method loses track in some sequences. ORB-SLAM3 (VO) means the VO front-end of ORB-SLAM3. RoMeO outperforms both learning-free and learning-based baselines significantly across the board. Both the metric-scale (ATE) and trajectory shape (RTE) improve by a large margin.

| Dataset | KITTI Odometry | 4Seasons | Cambridge | EuRoC | ETH3D | TUM-RGBD | Avg |
|---|---|---|---|---|---|---|---|
| ORB-SLAM3 (VO) | 29.10/163.25 | – | – | 0.488/1.677 | – | – | – |
| DSO | 47.23/154.25 | – | – | 0.404/1.532 | – | – | – |
| DeepV2d | 22.20/154.52 | 74.70/344.75 | 27.21/34.50 | 1.173/3.452 | 0.324/0.582 | 0.225/0.415 | 20.97/89.70 |
| TrianFlow | 42.07/168.17 | 40.44/150.46 | 27.35/34.60 | 1.731/1.848 | 0.706/0.868 | 0.444/0.565 | 18.79/59.41 |
| TartanVO | 32.25/45.30 | 59.25/81.70 | 19.40/24.23 | 0.632/4.135 | 0.421/26.273 | 0.320/16.692 | 18.71/33.06 |
| DROID-VO | 47.53/137.33 | 58.87/149.54 | 13.56/33.67 | 0.141/1.307 | 0.367/0.628 | 0.116/0.551 | 20.09/54.31 |
| DROID-Metric3d | 3.95/5.57 | 125.44/140.32 | 13.55/47.68 | 0.109/1.310 | 0.420/1.077 | 0.190/0.330 | 23.94/32.71 |
| DPVO | 46.04/140.28 | 9.95/141.36 | 15.89/36.03 | 0.101/1.865 | 0.203/0.646 | 0.107/0.324 | 12.05/53.42 |
| RoMeO-VO(Ours) | 2.71/3.81 | 19.59/42.56 | 9.96/23.47 | 0.098/1.126 | 0.022/0.238 | 0.067/0.091 | 5.40/11.88 |

- *4Seasons* (Wenzel et al., 2021): an outdoor driving dataset with diverse scenes and weather conditions.

- *Cambridge* (Kendall et al., 2015): a large-scale outdoor visual localization dataset taken around Cambridge University using a handheld camera.

- *EuRoC* (Burri et al., 2016): an indoor dataset captured by a Micro Aerial Vehicle (MAV).

- *TUM-RGBD* (Sturm et al., 2012): an indoor dataset captured with a handheld camera.

- *ETH3D* (Schops et al., 2019): we use the SLAM benchmark of ETH3D, which is an indoor dataset with LiDAR captured depth.

Since *4Seasons* and *ETH3D* contain too many sequences, we randomly select 1 training sequence of each scene for evaluation, see Appendix A for details.

**Metrics.** Following RGBD VO Campos et al. (2021), we evaluate different methods using the *Absolute Trajectory Error (ATE)*. To distinguish the metric scale accuracy and the trajectory shape accuracy, we define an additional metric where we align the scale of the output trajectory with the ground truth (GT); we call this metric *Relative Trajectory Error (RTE)*. Note that previous monocular VO papers Teed & Deng (2021); Teed et al. (2024) report RTE as ATE since they cannot recover metric scale poses and by default assume scale alignment with GT.

## 4.1 MAIN RESULTS

We first compare RoMeO with state-of-the-art methods. Since RoMeO can be used as a standalone VO system and also as part of a full SLAM pipeline, we conduct experiments for both applications.

Table 2 shows the comparison to VO systems. Learning-free baselines (ORB-SLAM3 (VO) and DSO) fail frequently on challenging data due to the instability of hand-crafted features. ORB-SLAM3 (VO) and DSO have $100\%$ success rates only on *KITTI Odometry* and *EuRoC*. Appendix A further reports results on individual sequences of each dataset. Learning-based baselines rarely lose track due to the improved stability of the end-to-end framework. However, the trajectory accuracy is limited especially on outdoor scenes. Most of them perform worse than ORB-SLAM3 (VO) on *KITTI Odometry*. The closest baseline to RoMeO is DROID-Metric3d, where Metric3d (Yin et al., 2023) depth is applied to the initialization and BA regularization of DROID-VO. Our implementation uses the original code from the Metric3d authors. The Metric3d paper showed significant performance improvement on *KITTI Odometry*, which was consistent with our experiment. However, the same strategy can hurt the performance on other datasets (e.g., RTE on *4Seasons* and *TUM-RGBD*, ATE on *Cambridge* and *ETH3D*).

RoMeO significantly and consistently outperforms both learning-based and learning-free baselines. Compared to the current SOTA DPVO, RoMeO reduces RTE and ATE respectively by $55.2\%$ and $77.8\%$ on average, and $> 90\%$ on challenging data, e.g., the ATE of *KITTI Odometry* improves from $140.28m$ to $3.81m$. This shows that RoMeO can significantly improve both the trajectory scale and shape, and generalizes to both indoor and outdoor scenes.

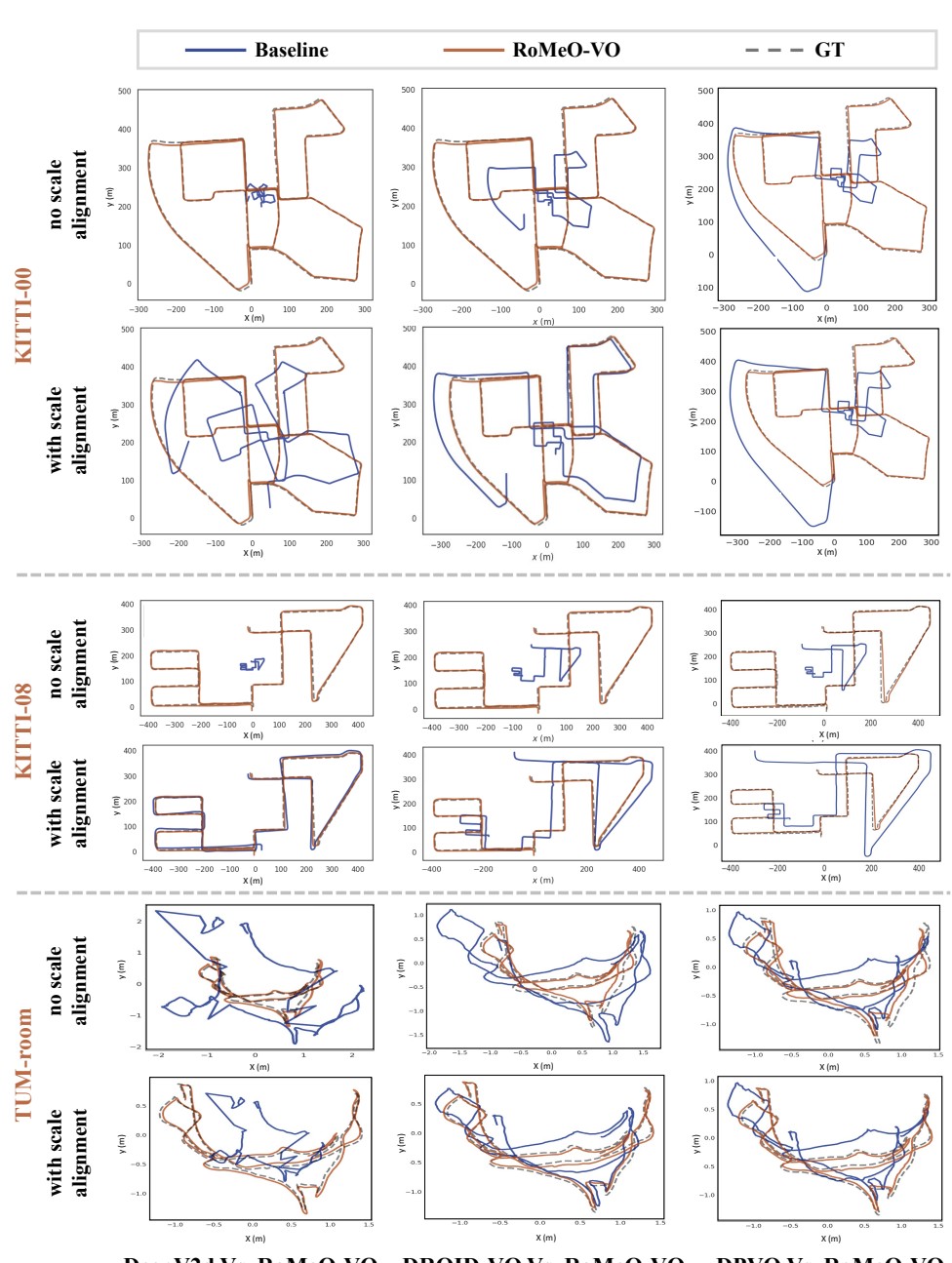

Figure 4: **Trajectory**. For each scene, the two rows show respectively the trajectory with and without scale alignment. RoMeO aligns much better with GT both with and without scale alignment.

Fig. 4 shows the trajectory visualizations. Consistent with the quantitative results, the predicted trajectory of RoMeO aligns better with GT both with and without scale alignment. Moreover, it is the only method that can close the loop without applying global bundle adjustment or loop closure. Fig. 5 shows the visualization of reconstructed point clouds. For both indoor and outdoor scenes, RoMeO provides dense reconstructions of much higher quality. The reconstruction of DPVO is extremely sparse since it applies sparse flow estimation to accelerate VO.

Table 3 shows the comparison to SLAM systems. We build RoMeO-SLAM with global bundle adjustment enabled. Similar to the case of VO, RoMeO effectively improves the accuracy of the full

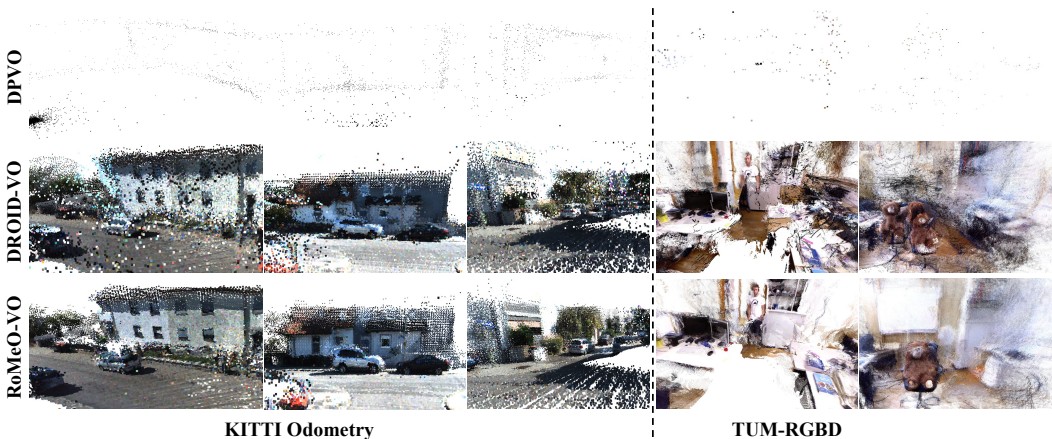

Figure 5: **Point cloud visualization**. *First two columns*: results on KITTI Odometry. *Last two columns*: results on TUM-RGBD. RoMeO provides dense and more accurate 3D reconstructions.

Table 3: **SLAM Evaluation**. Results are reported as RTE (m) (with scale alignment) / ATE (m). Missing values (–) indicate that the method loses track in some sequences. The performance gain of RoMeO on VO propagates to the full SLAM system.

| Dataset | KITTI odometry | 4Seasons | Cambridge | EuRoC | ETH3D | TUM-RGBD | Avg |
|---|---|---|---|---|---|---|---|
| ORB-SLAM3 | 16.42/165.28 | – | – | 0.214/1.437 | – | – | – |
| Droid-SLAM | 39.12/155.89 | 58.12/144.20 | 12.37/33.13 | 0.019/1.238 | 0.010/**0.011** | 0.028/0.168 | 18.28/55.77 |
| RoMeO-SLAM (Ours) | **2.64/3.72** | **20.39/44.20** | **9.90/23.23** | **0.016/1.100** | **0.008**/0.060 | **0.021/0.071** | **5.50/12.06** |

SLAM system. The improvement level remains similar as in the VO case, e.g., $93.3\%$ and $97.6\%$ reduction rate on *KITTI Odometry* for RTE and ATE respectively.

## 4.2 ANALYSIS

In this section, we provide a detailed method analysis. We first conduct ablation studies to validate each RoMeO component. Due to the computation cost, the ablation study is conducted on 3 evaluation datasets (1 indoor and 2 outdoor). The results are reported in Table 4.

**Depth guided BA.** To verify the effectiveness of our depth regularization in BA (E.q. 1), we report in row 2 (*always regularize depth*) and 3 (*no depth regularization*) of Table 4 the performance of RoMeO with depth regularization always enabled (whether the predicted depth is accurate or not) and always disabled. When the depth regularization is always enabled, the performance of RoMeO drops severely on *4Seasnos* ($19.59m \rightarrow 117.95m$ for RTE) due to the negative impact of noisy monocular/MVS depth. On the other hand, when the depth regularization is always disabled, the performance on *KITTI Odometry* drops heavily ($3.81m \rightarrow 47.91m$ for ATE) due to the lack of effective BA regularization. With adaptive noise filtering, RoMeO maintains most performance gain from depth regularization, and minimizes the negative impact from inaccurate monocular/MVS depth, performing robustly across different datasets.

Table 4: **Ablation study**. Removing or replacing RoMeO components hurts the performance (RTE/ATE). Noise augmented training is abbreviated as NAT.

| Model | Mono depth | NAT | MVS | KITTI Odometry | 4Seasons | TUM-RGBD | Mean RTE/ATE |
|---|---|---|---|---|---|---|---|
| RoMeO-VO (ours) | DPT | ✓ | ✓ | 2.71/3.81 | 19.59/42.56 | 0.067/0.091 | **7.45/15.49** |
| always regularize depth | DPT | ✓ | ✓ | 2.48/3.63 | 117.95/154.78 | 0.136/0.158 | 40.20/52.86 |
| no depth regularization | DPT | ✓ | ✓ | 11.25/47.91 | 16.99/39.63 | 0.058/0.088 | 9.44/29.21 |
| DPT → Metric3D | Metric3D | ✓ | ✓ | 3.23/4.67 | 53.43/89.28 | 0.135/0.288 | 18.94/31.41 |
| no MVS | DPT | ✓ | | 3.35/5.14 | 23.49/55.38 | 0.075/0.121 | 8.96/20.21 |
| no NAT & no MVS | DPT | | | 4.25/8.54 | 34.32/121.65 | 0.098/0.205 | 12.89/43.47 |
| no depth prior | | | | 47.53/137.33 | 58.87/149.54 | 0.116/0.551 | 35.52/95.79 |

**MVS prior.** RoMeO applies MVS depth to enhance the intermediate flow and pose estimation. As shown in row 5 of Table 4 (*no MVS*), removing MVS priors hurts both ATE and RTE, showing the importance of the proposed module.

**Noise augmented training.** Noise augmented training plays a crucial role in adapting the VO network to the depth-prior-enhanced inputs. As shown in row 6 of Table 4 (*no NAT & no MVS*), removing noise augmented training from *no MVS*, i.e., only using the pre-trained flow model in RoMeO, further worsens both ATE and RTE. E.g., in *4Seasons*, the ATE increases by $> 2$x. This result demonstrates the importance of fine-tuning to maximize the performance gain of RoMeO.

**Metric depth prior.** Depth initialization enables not only better flow initialization but also the ability to obtain accurate metric-scale poses. The last row of Table 4 (*no depth prior*) removes depth initialization from *no NAT & no MVR*, leading to a drastic error increase. On *KITTI Odometry*, both ATE and RTE increase by more than 10x, showing that the model cannot maintain a reasonable metric scale and accurate trajectory shape without depth initialization.

**Depth model compatibility.** RoMeO uses DPT-Hybrid as the monocular depth prior. To verify that RoMeO is compatible with other depth models, we apply the techniques of RoMeO to the Droid-Metric3D (Yin et al., 2023) baseline. We perform the same training and evaluation as for the DPT-based RoMeO except that the monocular depth model is changed to Metric3D. Comparing row 3 of Table 4 (*DPT → Metric3D*) with the Droid-Metric3D baseline in Table 2, we see that RoMeO is also compatible with other depth models and can effectively improve the general robustness. Meanwhile, changing DPT to Metric3D increases both ATE and RTE. This result shows the effectiveness of our depth model choice.

Table 5: **Efficiency of RoMeO and a fast variant.** The speed is measured on the same RTX-3090 GPU.

| Model | KITTI Odometry | | TUM-RGBD | |
|---|---|---|---|---|
| | RTE (m) / ATE (m) | FPS | RTE (m) / ATE (m) | FPS |
| No depth prior | 47.53/137.33 | 5.33 | 0.116/0.551 | 10.77 |
| RoMeO-VO | **2.71/3.81** | 3.54 | **0.067/0.091** | 7.65 |
| RoMeO-VO-fast | 4.23/6.89 | **6.28** | 0.077/0.140 | **20.57** |

**Efficiency.** RoMeO introduces two depth models to improve the performance and enable metric scale VO. Here we show that the overhead of introducing depth priors is small. We also demonstrate a fast version of RoMeO which is even faster than the base VO system without depth priors, while maintaining most performance gain. Tab. 5 compares the original and the fast version of RoMeO with the base architecture without depth priors. RoMeO-VO improves the accuracy significantly with marginal overhead. To create the fast version (RoMeO-VO-fast), we first reduce the input resolution, e.g., from 320*512 to 224*448 on KITTI Odometry and from 240*320 to 192*256 at TUM-RGBD. Then, we disable the monocular depth initialization on non-keyframes. To ensure the scale consistency of the non-keyframes, we use the depth of the nearest keyframe to initialize the non-keyframes. These simple strategies preserve most of the accuracy improvement while making RoMeO-VO-fast even faster than the base VO system without depth priors, achieving twice the speed of the base system on TUM-RGBD.

## 5 CONCLUSION

We propose *RoMeO*, a robust visual odometry (VO) system that can return metric-scale trajectories from monocular RGB videos without 3D sensors. RoMeO utilizes pre-trained monocular and multi-view depth models as effective priors for VO. It adaptively selects accurate depth outputs to regularize BA, creating a positive feedback loop that enhances optimization robustness against local minima. Noise-augmented training is introduced to fully adapt VO networks to depth-enhanced inputs while maintaining robustness to prior noise. RoMeO generalizes to both indoor and outdoor data. It consistently and significantly outperforms SOTA across 6 diverse zero-shot datasets. Besides improving the trajectory accuracy, RoMeO also enables more accurate 3D reconstructions. The performance gain also transfers to the full SLAM system. In terms of limitations, RoMeO currently uses separate models/hyperparameters for indoor and outdoor scenes. An interesting future direction would be to develop better depth estimation methods that are lightweight, and generalize to both indoor and outdoor scenes with a single model and hyperparameter setting. We also create a fast version of RoMeO by reducing the resolution and only initializing depth on keyframes. We believe these strategies can be further optimized with a more comprehensive study in the future.

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

# A APPENDIX

As shown in Tables 6 to 10, RoMeO outperforms the baselines on most individual sequences of each dataset. Unlike previous learning-based methods that often perform worse than conventional methods when they do not fail, RoMeO performs better and improves over SOTA on most sequences.

Table 6: **VO (top) and SLAM (bottom) results on the individual sequences of _KITTI Odometry_**(Geiger et al., 2012).

| VO result | 00 | 03 | 04 | 05 | 06 | 07 | 08 | 09 | 10 | average |
|---|---|---|---|---|---|---|---|---|---|---|
| ORB-SLAM3 (VO) | 49.43/182.37 | **0.64**/148.71 | 1.81/120.53 | 33.22/146.49 | 54.28/125.27 | 16.20/81.15 | 52.58/249.35 | 46.61/208.96 | 7.13/200.24 | 29.10/163.25 |
| DSO | 48.04/180.58 | 0.80/142.58 | **0.36**/73.46 | 48.45/143.82 | 57.59/136.59 | 53.67/66.80 | 113.02/230.02 | 92.19/218.11 | 11.03/196.32 | 47.23/154.25 |
| DeepV2d | 101.65/173.50 | 7.15/150.03 | 4.08/98.86 | 27.05/142.89 | 7.39/120.58 | 8.70/80.53 | 18.91/236.56 | 10.13/199.01 | 14.77/188.74 | 22.20/154.52 |
| TrianFlow | 96.43/186.91 | 10.26/163.78 | 6.05/109.07 | 52.69/154.90 | 39.00/132.37 | 15.69/88.18 | 56.63/256.00 | 72.59/218.89 | 29.35/203.44 | 42.07/168.17 |
| TartanVO | 63.84/63.91 | 7.71/21.56 | 2.89/49.45 | 54.61/56.57 | 24.67/47.10 | 19.29/19.95 | 59.55/59.56 | 32.61/59.23 | 25.04/30.41 | 32.25/45.30 |
| DROID-VO | 109.00/111.17 | 5.57/150.5 | 1.05/108.5 | 60.37/121.25 | 38.03/123.50 | 21.41/74.33 | 105.64/169.03 | 73.04/193.30 | 13.77/194.52 | 47.53/137.33 |
| DROID-Metric3d | **4.38/5.35** | 1.01/5.90 | 0.74/2.11 | 5.27/**7.72** | 3.19/3.76 | 8.84/11.27 | 6.59/8.09 | 2.99/3.00 | 2.55/2.97 | 3.95/5.57 |
| DPVO | 110.93/111.78 | 1.67/156.68 | 1.05/107.81 | 55.89/127.76 | 59.84/118.55 | 18.21/69.27 | 97.19/189.61 | 60.25/196.83 | 9.37/184.23 | 46.04/140.28 |
| RoMeO-VO (ours) | 4.78/5.88 | 0.98/**2.22** | 0.62/**1.77** | **3.35**/7.84 | **1.93/1.97** | **3.95/3.96** | **4.46/6.31** | **2.65/2.73** | **1.63/1.67** | **2.71/3.81** |

| SLAM result | 00 | 03 | 04 | 05 | 06 | 07 | 08 | 09 | 10 | average |
|---|---|---|---|---|---|---|---|---|---|---|
| ORB-SLAM3 | 9.27/189.54 | **0.58**/144.83 | 1.69/119.27 | 5.73/162.10 | 16.21/136.77 | 2.71/83.21 | 51.01/246.46 | 53.70/210.03 | 6.88/195.34 | 16.42/165.28 |
| Droid-SLAM | 74.41/167.26 | 7.37/156.09 | **0.43**/105.63 | 60.14/140.15 | 38.45/126.85 | 20.66/81.38 | 69.33/228.68 | 65.81/207.15 | 15.48/189.84 | 39.12/155.89 |
| RoMeO-SLAM (ours) | **4.74/5.75** | 0.95/**2.18** | 0.57/**1.76** | **3.28/7.28** | **1.88/1.90** | 3.55/**3.78** | **4.52/6.45** | **2.63/2.69** | **1.65/1.71** | **2.64/3.72** |

Table 7: **VO (top) and SLAM (bottom) results on the individual sequences of _4Seasons_** (Wenzel et al., 2021).

| VO result | business_campus | old_town | parking_garage | neighborhood | office_loop | city_loop | countryside | average |
|---|---|---|---|---|---|---|---|---|
| ORB-SLAM3 (VO) | 88.21/163.57 | 15.10/201.25 | 4.58/18.54 | 74.95/160.62 | 48.32/80.13 | – | – | – |
| DSO | 143.23/163.79 | – | – | – | 72.52/166.32 | 56.31/69.72 | 56.72/102.61 | – |
| DeepV2d | 62.03/1547.58 | 36.01/204.19 | 18.81/19.06 | 98.33/129.06 | 88.50/173.16 | 105.75/137.01 | 113.49/203.21 | 74.70/344.75 |
| TrianFlow | 20.92/161.39 | 37.05/216.35 | 13.29/19.02 | 27.80/143.48 | 31.94/172.04 | 63.32/116.43 | 88.75/224.53 | 40.44/150.46 |
| TartanVO | 80.37/133.28 | 69.18/69.93 | 10.69/37.92 | 88.83/136.69 | 68.75/68.81 | 17.69/**31.23** | 79.23/94.03 | 59.25/81.70 |
| DROID-VO | 23.06/161.54 | 8.85/209.24 | 14.32/19.15 | 12.19/143.04 | 60.71/171.98 | 115.03/116.57 | 177.93/225.24 | 58.87/149.54 |
| DROID-Metric3d | 94.33/104.09 | 212.85/217.34 | 12.92/13.78 | 98.37/135.31 | 142.19/170.17 | 96.77/116.40 | 220.63/225.13 | 125.44/140.32 |
| DPVO | 8.19/161.27 | 19.68/161.88 | **0.98/14.89** | **7.68**/140.77 | **21.54**/170.91 | **1.96**/115.41 | **9.63**/224.38 | **9.95**/141.36 |
| RoMeO-VO (ours) | **7.41/13.74** | **3.97/27.87** | 10.86/35.60 | 11.89/**18.13** | 38.78/**72.22** | 24.59/77.69 | 39.62/**52.67** | 19.59/**42.56** |

| SLAM result | business_campus | old_town | parking_garage | neighborhood | office_loop | city_loop | countryside | average |
|---|---|---|---|---|---|---|---|---|
| ORB-SLAM3 | 100.35/163.21 | 14.02/206.53 | **2.79/16.83** | 78.82/163.25 | 48.16/82.07 | – | – | – |
| Droid-SLAM | 23.55/152.99 | 8.54/205.85 | 14.38/16.40 | 14.28/134.05 | 59.04/158.63 | 113.54/116.52 | 173.50/224.93 | 58.12/144.20 |
| RoMeO-SLAM (ours) | **8.32/13.84** | **4.04/27.49** | 10.51/36.03 | **11.59/20.49** | 42.05/77.89 | 26.87/80.26 | 39.37/53.45 | 20.39/44.20 |

Table 8: **VO (top) and SLAM (bottom) results on the individual sequences of _Cambridge_** (Kendall et al., 2015).

| VO result | GreatCourt | KingsCollege | OldHospital | ShopFacade | StMarysChurch | Street | average |
|---|---|---|---|---|---|---|---|
| ORB-SLAM3 (VO) | – | – | – | – | – | – | – |
| DSO | – | – | – | – | – | – | – |
| DeepV2d | 27.55/33.38 | 15.27/29.24 | 8.17/10.78 | 7.18/8.73 | 12.04/15.75 | 93.06/109.12 | 27.21/34.50 |
| TrianFlow | 27.93/32.93 | 21.42/30.15 | 7.30/10.84 | 5.97/8.96 | 11.93/16.05 | 89.52/108.67 | 27.35/34.60 |
| TartanVO | 28.09/34.31 | 12.81/16.92 | 6.60/7.54 | 4.01/4.22 | 10.60/11.82 | 54.28/**70.60** | 19.40/24.23 |
| DROID-VO | 14.79/32.48 | 1.11/29.2 | **0.94**/11.92 | 0.53/8.09 | 4.35/14.99 | 59.66/105.34 | 13.56/33.67 |
| DROID-Metric3d | 13.87/40.80 | 0.81/18.08 | 1.31/**7.53** | 0.75/12.98 | 4.41/21.51 | 60.16/185.23 | 13.55/47.68 |
| DPVO | 37.28/38.34 | **0.18/16.12** | 4.07/10.86 | **0.12**/8.68 | 5.27/18.09 | 48.39/124.06 | 15.89/36.03 |
| RoMeO-VO (ours) | **12.78/26.79** | 0.40/19.2 | 1.14//8.56 | 0.52/**3.27** | **4.28/10.28** | 40.68/72.76 | **9.96/23.47** |

| SLAM result | GreatCourt | KingsCollege | OldHospital | ShopFacade | StMarysChurch | Street | average |
|---|---|---|---|---|---|---|---|
| ORB-SLAM3 | – | – | – | – | – | – | – |
| Droid-SLAM | 13.78/32.49 | 0.34/29.23 | **0.89**/10.58 | 0.51/8.25 | **4.17**/15.14 | 54.50/103.08 | 12.37/33.13 |
| RoMeO-SLAM (ours) | **12.70/26.73** | **0.30/18.70** | 1.06/**8.44** | **0.49/3.23** | 4.25/**10.35** | **40.61/69.66** | **9.90/23.23** |

Table 9: **VO (top) and SLAM (bottom) results on the individual sequences of *EuRoC* (Burri et al., 2016).**

| VO result | V101 | V102 | V103 | V201 | V202 | V203 | average |
|---|---|---|---|---|---|---|---|
| ORB-SLAM3 (VO) | **0.036**/1.045 | 0.139/4.210 | 0.713/1.108 | 1.352/1.866 | **0.047**/0.302 | 0.642/1.529 | 0.488/1.677 |
| DSO | 0.089/0.937 | **0.107**/3.859 | 0.903/1.236 | 0.044/1.244 | 0.132/0.361 | 1.152/1.555 | 0.404/1.532 |
| DeepV2d | 0.717/1.365 | 0.695/6.210 | 1.483/5.544 | 0.839/3.109 | 1.052/2.331 | 0.591/2.153 | 1.173/3.452 |
| TrianFlow | 0.895/1.243 | 3.956/4.038 | 0.974/1.076 | 1.849/1.927 | 0.483/0.547 | 2.229/2.257 | 1.731/1.848 |
| TartanVO | 0.447/2.357 | 0.389/6.901 | 0.622/5.921 | 0.433/3.644 | 0.749/3.015 | 1.152/2.973 | 0.632/4.135 |
| DROID-VO | 0.103/1.203 | 0.165/1.847 | 0.158/1.061 | 0.102/1.360 | 0.115/0.992 | 0.204/1.378 | 0.141/1.307 |
| DROID-Metric3d | 0.054/1.234 | 0.141/**1.815** | 0.112/1.135 | 0.075/1.253 | 0.116/1.057 | 0.156/1.366 | 0.109/1.310 |
| DPVO | 0.048/1.190 | 0.148/3.658 | **0.093/0.759** | 0.059/1.626 | 0.051/2.361 | 0.207/1.598 | 0.101/1.865 |
| RoMeO-VO (ours) | 0.073/**0.892** | 0.136/2.634 | 0.101/0.991 | **0.056/0.970** | 0.107/**0.132** | **0.117/1.139** | **0.098/1.126** |

| SLAM result | GreatCourt | KingsCollege | OldHospital | ShopFacade | StMarysChurch | Street | average |
|---|---|---|---|---|---|---|---|
| ORB-SLAM3 | 0.033/0.992 | 0.042/3.854 | 0.395/0.952 | 0.683/1.705 | 0.028/0.282 | 0.103/0.837 | 0.214/1.437 |
| Droid-SLAM | 0.037/0.920 | **0.013/1.430** | 0.019/1.123 | 0.017/1.437 | 0.014/1.100 | **0.013**/1.418 | 0.019/1.238 |
| RoMeO-SLAM (ours) | **0.031/0.88** | **0.013**/2.556 | **0.018/0.990** | **0.016/0.963** | **0.013/0.111** | **0.013**/1.118 | **0.016/1.100** |

Table 10: **VO (top) and SLAM (bottom) results on the individual sequences of *ETH3D SLAM* (Schops et al., 2019).**

| VO result | cables | camera_shake | desk_changing | einstein | planar | mannequin_face | sfm_lab_room | average |
|---|---|---|---|---|---|---|---|---|
| ORB-SLAM3 (VO) | – | – | – | – | 0.021/**0.088** | 0.535/0.539 | – | – |
| DSO | – | – | 1.312/1.530 | – | 0.686/0.689 | 0.490/0.491 | – | – |
| DeepV2d | 0.172/0.130 | 0.155/1.394 | 0.943/1.065 | 0.407/0.467 | 0.424/0.430 | 0.119/**0.124** | 0.051/0.466 | 0.324/0.582 |
| TrianFlow | 0.262/0.601 | 0.152/0.206 | 1.373/1.485 | 1.037/1.044 | 0.729/0.730 | 0.447/0.475 | 0.941/1.535 | 0.706/0.868 |
| TartanVO | 0.132/24.768 | 0.152/9.070 | 0.851/51.524 | 0.831/33.775 | 0.506/27.802 | 0.197/17.822 | 0.277/19.152 | 0.421/26.273 |
| DROID-VO | **0.013**/0.261 | 0.140/0.194 | 0.186/1.090 | 0.824/0.876 | 0.010/0.126 | 0.329/0.366 | 1.066/1.485 | 0.367/0.628 |
| DROID-Metric3d | 0.262/0.364 | 0.137/0.525 | 1.139/2.668 | 0.612/1.023 | 0.354/0.492 | 0.290/2.139 | 0.146/0.328 | 0.420/1.077 |
| DPVO | 0.020/0.551 | 0.142/0.332 | 1.192/2.383 | 0.022/0.358 | 0.022/0.289 | 0.003/0.134 | **0.021**/0.479 | 0.203/0.646 |
| RoMeO-VO (ours) | 0.027/**0.042** | **0.048/0.131** | 0.028/0.396 | **0.008/0.340** | **0.006**/0.550 | **0.002**/0.158 | 0.036/**0.050** | **0.022/0.238** |

| SLAM result | cables | camera_shake | desk_changing | einstein | planar | mannequin_face | sfm_lab_room | average |
|---|---|---|---|---|---|---|---|---|
| ORB-SLAM3 | – | – | – | – | 0.012/0.224 | 0.287/0.335 | – | – |
| Droid-SLAM | **0.005/0.007** | 0.049/0.049 | **0.004/0.005** | 0.004/**0.005** | **0.002/0.002** | **0.002/0.004** | **0.005/0.008** | 0.010/**0.011** |
| RoMeO-SLAM (ours) | 0.017/0.028 | **0.015/0.032** | **0.004**/0.118 | **0.002**/0.108 | **0.002**/0.004 | **0.002**/0.060 | 0.016/0.070 | **0.008**/0.060 |

Table 11: **VO (top) and SLAM (bottom) results on the individual sequences of *TUM-RGBD* (Sturm et al., 2012).**

| VO result | 360 | desk | desk2 | floor | plant | room | rpy | teddy | xyz | average |
|---|---|---|---|---|---|---|---|---|---|---|
| ORB-SLAM3 (VO) | – | **0.017**/0.065 | – | – | **0.034**/0.492 | – | – | – | 0.009/0.061 | – |
| DSO | – | 0.405/0.535 | 0.322/0.818 | **0.041**/0.119 | 0.108/0.302 | 0.800/0.850 | – | – | 0.058/0.073 | – |
| DeepV2d | 0.144/0.250 | 0.105/0.327 | 0.321/0.578 | 0.628/1.514 | 0.217/0.330 | 0.215/0.250 | 0.046/0.115 | 0.294/0.312 | 0.051/0.055 | 0.225/0.415 |
| TrianFlow | 0.187/0.187 | 0.526/0.698 | 0.483/0.764 | 0.739/0.741 | 0.388/0.610 | 0.884/0.929 | 0.050/0.158 | 0.554/0.762 | 0.182/0.236 | 0.444/0.565 |
| TartanVO | 0.160/7.212 | 0.478/23.082 | 0.539/25.080 | 0.348/20.811 | 0.395/10.374 | 0.417/19.490 | 0.050/10.289 | 0.329/19.791 | 0.160/14.095 | 0.320/16.692 |
| DROID-VO | 0.141/0.556 | 0.064/0.335 | 0.078/1.001 | 0.063/0.152 | 0.041/0.631 | 0.393/0.704 | 0.030/0.811 | 0.221/0.714 | 0.017/0.057 | 0.116/0.551 |
| DROID-Metric3d | 0.222/0.263 | 0.139/0.260 | 0.075/0.109 | 0.195/0.769 | 0.271/0.470 | 0.264/0.378 | 0.019/**0.019** | 0.507/0.669 | 0.019/0.036 | 0.190/0.330 |
| DPVO | 0.156/0.174 | 0.034/0.258 | 0.050/0.216 | 0.183/0.270 | 0.034/0.575 | 0.383/0.400 | 0.038/0.104 | 0.073/0.852 | 0.012/0.069 | 0.107/0.324 |
| RoMeO-VO (ours) | **0.084/0.085** | 0.049/**0.051** | **0.038/0.059** | 0.103/**0.104** | 0.079/0.133 | **0.167/0.169** | **0.015**/0.030 | **0.069**/0.217 | **0.005/0.006** | **0.067/0.091** |

| SLAM result | 360 | desk | desk2 | floor | plant | room | rpy | teddy | xyz | average |
|---|---|---|---|---|---|---|---|---|---|---|
| ORB-SLAM3 | – | 0.016/0.069 | – | – | 0.038/0.510 | – | – | – | 0.145/0.728 | **0.005**/0.087 | – |
| Droid-SLAM | 0.063/0.081 | 0.017/0.100 | **0.026**/0.152 | **0.022/0.055** | 0.016/0.291 | 0.043/0.280 | 0.023/0.023 | 0.035/0.515 | 0.010/0.022 | 0.028/0.168 |
| RoMeO-SLAM (ours) | **0.038/0.038** | **0.012/0.083** | 0.027/0.080 | 0.033/0.105 | **0.011/0.014** | **0.035/0.083** | **0.012/0.019** | 0.020/0.213 | **0.005/0.005** | **0.021/0.071** |

