# OpenReview forum: "ROMEO: ROBUST METRIC VISUAL ODOMETRY"
_ICLR.cc/2025/Conference — ICLR 2025 Conference Withdrawn Submission_

### Official Review · Reviewer_Ws6N · 2024-10-25

**Soundness:** 2
**Presentation:** 2
**Contribution:** 2
**Rating:** 3
**Confidence:** 5

**Summary:**

This system introduces a new learning-based visual odometry approach, RoMeO, which utilizes lightweight monocular depth models to initialize and recover metric scale. By integrating robust depth-guided bundle adjustment, the approach adaptively employs monocular and multi-view stereo priors to optimize pose-depth relationships. RoMeO implements a strategy to mitigate the impact of depth noise by comparing the photometric error between adjacent keyframes to determine the appropriate usage of depth priors. Additionally, noise-augmented training enhances accuracy while maintaining resilience to uncertainties in depth prediction.

**Strengths:**

1)	The proposed method demonstrates superior performance compared to state-of-the-art methods across various benchmarks.

**Weaknesses:**

1)	The technical contribution and insights appear limited. The integration of additional depth estimation models to enhance the performance of monocular VO or SLAM systems has become a well-recognized and practical solution. The primary distinction of this paper compared to previous literature is the incorporation of both monocular depth estimation models and multi-view stereo (MVS) models.
2)	The paper presents a less rigorous concept that could potentially mislead the community. The authors claim that existing monocular VO approaches lack robustness and fail to generalize to unseen data, particularly in recovering metric-scale poses. However, this paper overstates its contributions, as it relies on predefined scale and shift hyperparameters to convert predictions from the DPT model to metric or absolute scale. These parameters must be computed using known ground truth to achieve reasonable performance across different benchmarks, rendering the method neither robust nor generalizable to unknown data.
3)	The operational speed of this system (3-20 fps) is significantly slower compared to the baseline DPVO, which claims to achieve speeds of 60-120 fps.
4)	The schematic diagram, specifically Figure 2, is cluttered, making it difficult to discern the data flow for training and inference.

**Questions:**

How can one determine the scale and shift hyperparameters required to enforce DPT model predictions into metric scale in an unseen environment?

---

### Official Review · Reviewer_3Qtm · 2024-10-29

**Soundness:** 3
**Presentation:** 2
**Contribution:** 3
**Rating:** 6
**Confidence:** 3

**Summary:**

The work proposes a novel visual odometry system, RoMeO, that incorporates metric depth, MVS depth, and flow estimates to supervise differentiable Bundle Adjustment. Quantitative and qualitative results demonstrate the superiority of camera tracking across 6 benchmarks. Further ablation studies validate the design choices. Analysis of various depth estimators provides insights into the appropriate supervision for visual odometry systems.

**Strengths:**

1. The paper shows superior camera tracking performance against learning-free and learning-based VO methods. As part of the SLAM system, it achieves the best performance compared to DROID-SLAM and ORB-SLAM3 approaches.
2. The paper is well-written, the experiments are extensive, and the superiority of the proposed system in camera tracking is demonstrated across several benchmarks.
3. Analysis of different depth estimators provides interesting insights into the right supervision for visual odometry systems.

**Weaknesses:**

1. The pipeline flow is not completely clear from the method description and Figure 2. It would be beneficial to separate the trained or frozen components (Depth, Flow, MVS Networks) from the test-time optimization (differentiable BA, sliding window optimization) in the figure and the text. A figure for the Depth-guided BA (Section 3.1) seems necessary to understand the optimization procedure.
2. No demo is provided to demonstrate the proposed system's pipeline step-by-step and performance.

**Questions:**

1. What is the residual flow that is estimated by the flow network? Is it a correction for inaccurate pose and depth?
2. How does RoMeO perform against D3VO [1] on the KITTI and EuRoC benchmarks as per [1]? Based on their selected sequences, RoMeO is less accurate than D3VO; therefore, the evaluation with D3VO is important to confirm the SOTA claims.
3. How accurate is the estimated depth? It would be beneficial if the depth evaluation of the estimates from DPT-Hybrid and BA is provided (KITTI Eigen [2], EuRoC MAV). Although it is claimed that the system is robust with respect to noisy depth (contribution 3), it seems that depth initialization should be quite good already since the number of iterations until convergence of BA is rather small (2 iterations each time).
4. What optimization method is used for BA?
5. How is C$_{i}$ (Eq. 1) "automatically" estimated based on the amount of noise in $\boldsymbol{d}^{*}_i$?
6. While the ablation studies show a positive impact on pose accuracy, what is the rationale behind combining monocular and MVS depth estimators? Do they provide a complementary supervision signal for VO?
7. Is the system applicable for embedded devices?

[1] Yang et al. "D3vo: Deep depth, deep pose and deep uncertainty for monocular visual odometry.", 2020.
[2] Eigen et al. "Depth map prediction from a single image using a multi-scale deep network.", 2014.

---

### Official Review · Reviewer_2xGu · 2024-11-03

**Soundness:** 3
**Presentation:** 3
**Contribution:** 2
**Rating:** 6
**Confidence:** 4

**Summary:**

This paper presents a monocular RGB visual odometry (VO) algorithm that maintains high accuracy in outdoor scene even without input from 3D sensors like depth cameras or LiDAR. Specifically, the authors build on the deep visual odometry framework by Teed & Deng (2021). They introduce pre-trained monocular depth and multi-view stereo neural networks to provide frame-by-frame depth priors. These priors serve two key roles: 1) as initialization for subsequent networks, and 2) as regularization terms for bundle adjustment (BA) optimization. The algorithm is tested across three indoor and three outdoor datasets. Experimental results demonstrate that incorporating depth priors significantly improves the scale estimation of the visual odometry, especially in outdoor datasets where the method achieves remarkable accuracy. The authors also conduct detailed ablation studies, verifying the necessity of their proposed approach.

**Strengths:**

* The experimental results in this paper are thorough. The authors use the TartanAir dataset for pre-training and conduct evaluations on three indoor and three outdoor datasets. The results are consistent and achieve state-of-the-art (SOTA) accuracy on most datasets. Notably, the proposed method shows substantial improvements in outdoor datasets, which strongly supports the effectiveness of the approach.

* The results of this paper have practical implications. Theoretically, monocular RGB VO is constrained to small-scale scenes due to the lack of absolute scale observation. Additionally, using monocular depth estimation networks as ground truth can introduce biases due to noise and generalization issues, potentially degrading VO performance. However, the impressive performance on the KITTI dataset demonstrates that monocular depth estimation networks can be used as a regularization tool for scale, providing globally consistent scale information. This finding could be valuable for the SLAM community.

**Weaknesses:**

* The paper's motivation is questionable. Although the results are insightful, the motivation is not sufficiently compelling. The authors claim that introducing pre-trained monocular depth and multi-view stereo neural networks helps recover scene scale. However, the scale of each frame is merely a scalar, and using dense depth priors for this purpose significantly increases the system's computational load (with system efficiency below 30 FPS, making it non-real-time). A more elegant, well-designed "scale estimation network" might achieve similar results with less overhead.

* The discussion in this paper is lacking as follows:
  1) The reasons why pre-trained monocular depth and multi-view stereo neural networks aid in scale recovery are not fully explored. It could be due to the proposed strategy or the inherent capabilities of the networks used. Although some intuitive explanations are provided in the ablation study section, more extensive experimental evidence is needed;
  2) The introduction of pre-trained monocular depth and multi-view stereo neural networks did not significantly improve accuracy in indoor scenes. This suggests that in scenarios with good initial values and minimal scale variation, depth priors may not substantially enhance accuracy. This might be due to the difficulty of obtaining good initial values in outdoor scenes or rapid scale changes;
  3) The upper performance limits of the algorithm should be explored, such as providing ground-truth initialization or absolute scale input for comparison methods;
  4) The efficiency of the algorithm varies notably between indoor and outdoor environments. In Table 5, RoMeO-VO achieves FPS values of 3.54 on the KITTI dataset and 7.65 on the TUM-RGBD dataset, well beyond the range influenced by noise, yet the authors do not address this discrepancy.

**Questions:**

* Line 232: The terms "indoor" and "outdoor" are reversed.
* The flowchart in Figure 2 needs refinement. The current version does not convey the overall method clearly. For example, terms like "condition filter" and "adaptive noise filter" are not mentioned in the text, the meaning of arrows with different colors is unclear, and "projection" is not explained.

---

### Official Review · Reviewer_zf2w · 2024-11-03

**Soundness:** 2
**Presentation:** 3
**Contribution:** 2
**Rating:** 5
**Confidence:** 4

**Summary:**

The paper targets a monocular visual odometry problem with metric scale recovery, one critical problem in the SLAM community. Building upon the DROID-SLAM system, the paper shows that by exploiting a pre-trained and noisy metric depth estimation model, robust pose and depth can be obtained on multiple benchmarks in both indoor (EuRoC, TUM-RGBD, ETH3D) and outdoor(KITTI, 4Season, Cambridge) environments.

The optimization objective is the same as the RGB-D version of DROID-SLAM (eq. 1), where monocular depth estimation results (DPT or Metric3D) and MVS estimation results (MaGNet) are taken as the depth inputs. The authors try different tricks to handle the noisy depth estimation, where the role of each module is elaborated in the ablation study (Tab. 4).

Generally, it seems like a technical report to me. The system is not novel as mentioned in the claimed contributions, and the analysis is not sufficient and in-depth enough to point out the major issues in existing monocular and MVS depth models. The results are interesting and detailed, but how the proposed method assures 'robustness' and 'metric scale' (and the corresponding contribution to the community) lacks a clear and unified demonstration. The authors are encouraged to take a closer look at the results for a better understanding of the robust metric visual odometry problem.

**Strengths:**

+ The proposed method leads to improved accuracy of the visual odometry, especially in outdoor scenes where DROID_SLAM performs badly.
+ The adopted strategies bring improvement compared to the naive solution of DROID_Metric3d baseline.
+ Thorough experiments are conducted on diverse datasets, and the proposed method achieves the best results consistently.

**Weaknesses:**

The major concern is that the paper in the existing form provides little knowledge advances beyond better results. I don't doubt its efficacy on these evaluated datasets, but the straightforward solution requires better analysis to uncover the major issues regarding leveraging a pre-trained depth estimation model:
+ What are the typical failure modes on diverse datasets with a naive DROID-Metric3D baseline?
+ Is there any domain gap between different datasets?
+ What are the differences between monocular depth estimation and MVS results?
+ As there is no regularization term regarding the scale consistency, is the metric scale provided by Metric3D or DPT good enough to be ignored?

Besides the understanding of the robust metric visual odometry problem, the paper lacks sufficient understanding of its own strategies. All strategies to ensure 'robustness' rely heavily on hyperparameters ($\alpha$ in L207, Eq. 3, 20% in L279, small learning rate in L287, parameters in L310 to switch between indoor and outdoor scenes). It is unclear if the system is sensitive to these parameters. Besides, several arguments are not convincing to me:
+ Why depth regularization is not enabled during the initialization (L194)?
+ Why better depth estimation models lead to worse results as in Tab. 1 and Tab. 4 (DPT $\rightarrow$ Metric3D)? I don't see any relevance between the accuracy and the overhead as the system does not reduce the number of frames for depth estimation to balance the efficiency in this experiment.
+ What about the computational cost of MVS depth estimation?
+ Sec. 3.4 is not well validated. There is no evidence that 'purposely maintaining noise' (L284) leads to better robustness. It is also weird to ignore MVS depth during fine-tuning to 'ensure sufficient depth noise' (L288).
+ More trajectory results in different datasets should be provided to understand how other baselines fail. Video demonstration of the online dense reconstruction may better help the readers understand the major challenges of the 'robust metric visual odometry'.

Without a clear analysis of missing pieces toward robust metric visual odometry', the paper fails to address the motivations behind its system design, but instead adopts a simple fashion to only leverage reliable (heuristically determined) depth results for depth regularization. The major improvement regarding the pose estimation is on outdoor datasets where DROID-SLAM performs badly, where the robustness in indoor environments due to the proposed method is not clearly presented. The current form is not sufficient enough to be spread in the ICLR community. I list several suggestions in the following [Questions] section so the authors can improve the paper's quality with a more in-depth analysis.

**Questions:**

The paper ignores the regularization of noisy depth estimates in several cases: 1) Initialization; 2) large average photometric error; 3) large motion and lack of view intersections (MVS prior); 4) depth estimates with low confidence (MVS prior). The paper can be better polished given the following modifications besides the abovementioned problems:
+ Replacing the hard-coded strategies with more adaptive robust kernel functions to remove the outliers, or at least analyze the sensitivity of the system to the hyperparameters.
+ Justifying the differences in depth quality given monocular and MVS depth estimation. (Besides, does MaGNet provide metric depth estimates?) It should be more sufficient to see the gap among depth sensors, metric monocular depth estimates, and MVS depth estimates.
+ Presenting the typical failure mode with different monocular depth models in Tab. 1
+ The residuals are lower than the baseline given 3 iterations as in Fig. 3. It seems that the method provides the figure without leveraging its early convergence characteristic.
+ The benefits from this fine-tuning stage (Sec. 3.4) and the relevant key factors should be specified and highlighted so the irrelevant details can be ignored if it does not change the final results. What is the accuracy without NAT but with MVS (complementary to Tab. 4).
+ More experiments regarding 4Seasons and Cambridge should be provided as there is a large performance gap between the proposed method and the baselines. How DROID-Metric3D fails should be specifically demonstrated.
+ What if we only disable the monocular depth initialization on non-keyframes but do not change the image resolution (as in Tab. 5)?

---

### Official Review · Reviewer_4FGs · 2024-11-04

**Soundness:** 3
**Presentation:** 3
**Contribution:** 3
**Rating:** 1
**Confidence:** 5

**Summary:**

This paper proposes to bring in monocular depth estimates to avoid scale drift in visual odometry and shows that this works well (not too surprising).

**Strengths:**

The method seems to make reasonable use of monocular depth estimates to improve visual odometry (but results are not as clear given results pollution with KITTI overfitting, see below).

**Weaknesses:**

main issue:
Several of the comparisons are problematic.  The teaser compares to a method that does not estimate absolute scale (Fig.1 (b) right-side) and the proposed method is explicitly overfitted to KITTI data (uses a depth method which was explicitly overfitted to KITTI, which is particularly problematic for a driving setting with a fixed camera-road configuration throughout the whole dataset which allows almost trivially perfect scale estimates, especially for the road in front of the vehicle).  Notice that the better absolute depth can also help relative pose estimates as it provides consistent estimates across views, so this might not only affect ATE, but also RTE errors.  Therefore, Fig. 1 (a) should be recomputed excluding any KITTI data.  Similarly Fig. 4 is fundamentally problematic due to the KITTI overfitting issue (of DPT).  All Tables should excluded KITTI data, and have recomputed averages.  The overfitting can directly be seen from e.g. Table 2 which still has good performance for 4Seasons and Cambridge, but nowhere close as good as on Kitty.  Kitty odometry results should NOT be used given the explicit overfitting in DPT (with outdoor hyperparameters), and thus also not be included in any average.    Also, separate averages should be computed between indoor and outdoor datasets given the huge difference in scale (otherwise results are dominated by outdoor errors).

No ATE values should be provided for methods that do not attempt to compute metric scale!

other issues:
Equation (1) does not seem to be invariant to the overall scale of the observed scene with the first term being photometric and the second term proportional to absolute depth.  It is suprising that this works well on indoor and outdoor datasets, but might be thanks to C_i learning to switch off predictions that are further away (although they would carry useful information).

Also problematic is that different between absolute depth is used as an error, while for this typically inverse depth is much better (as the depth uncertainty from images is proportional to inverse depth or inverse squared depth depending on the scenarios).

The authors incorrectly equate their uncertainty weighting to Mahalanobis distance.  Also, it is not clear why monocular depth estimates are also weighted by uncertainty on flow, given they should be independent.

It was not clear how the initial absolute depth estimate is obtained given l.194 stating that for initialization depth priors are turned off.  Assuming the authors do incorporate depth priors, this might need clarification/reformulation to make this part more clear.

l. 232 indoor KITTI and outdoor TUM is swapped.
l. 249 0.1m is an absolute threshold which is unlikely to generalize well between indoor wearable and outdoor vehicles, as well as drones, etc.  A metric that is proportional to observed parallax would make more sense.  It is also not clear why the authors would turn off MVS guidance when the camera travels straight...  This makes no sense. Notice this would actually turn of MVS guidance for most of KITTI and other outdoor datasets (driving down straight roads).

**Questions:**

See weaknesses above.

**Details Of Ethics Concerns:**

The use of KITTI as explained above is highly problematic.  Not sure if the authors realized the issue, but it definitely is an issue.  Also showing that methods that don't estimate scale are bad at estimating scale is also inappropriate.

---

### Note · Authors · 2024-11-25

I have read and agree with the venue's withdrawal policy on behalf of myself and my co-authors.